# Fabrication and Characterization of Highly Efficient As-Synthesized WO_3_/Graphitic-C_3_N_4_ Nanocomposite for Photocatalytic Degradation of Organic Compounds

**DOI:** 10.3390/ma15072482

**Published:** 2022-03-28

**Authors:** Mai S. A. Hussien, Abdelfatteh Bouzidi, Hisham S. M. Abd-Rabboh, Ibrahim S. Yahia, Heba Y. Zahran, Mohamed Sh. Abdel-wahab, Walaa Alharbi, Nasser S. Awwad, Medhat A. Ibrahim

**Affiliations:** 1Department of Chemistry, Faculty of Education, Ain Shams University, Roxy, Cairo 11757, Egypt; maisalehamar@gmail.com; 2Nanoscience Laboratory for Environmental and Bio-Medical Applications (NLEBA), Department of Physics, Faculty of Education, Ain Shams University, Roxy, Cairo 11757, Egypt; 3Laboratory of Materials for Energy and Environment, and Modeling (LMEEM), Faculty of Sciences of Sfax, University of Sfax, B.P. 1171, Sfax 3038, Tunisia; bouzidi.abdelfatteh@yahoo.fr; 4Department of Chemistry, Faculty of Science, King Khalid University, P.O. Box 9004, Abha 61421, Saudi Arabia; habdrabboh@kku.edu.sa (H.S.M.A.-R.); nsawwad20@yahoo.com (N.S.A.); 5Department of Chemistry, Faculty of Science, Ain Shams University, Abbassia, Cairo 11566, Egypt; 6Laboratory of Nano-Smart Materials for Science and Technology (LNSMST), Department of Physics, Faculty of Science, King Khalid University, P.O. Box 9004, Abha 61413, Saudi Arabia; dr_isyahia@yahoo.com (I.S.Y.); dr_hyzahran@yahoo.com (H.Y.Z.); 7Research Center for Advanced Materials Science (RCAMS), King Khalid University, P.O. Box 9004, Abha 61413, Saudi Arabia; 8Semicondcuotr Laboratory, Department of Physics, Faculty of Education, Ain Shams University, Roxy, Cairo 11757, Egypt; 9Materials Science and Nanotechnology Department, Faculty of Postgraduate Studies for Advanced Sciences, Beni-Suef University, Beni-Suef 62511, Egypt; 10Department of Chemistry, Science and Arts College, Rabigh Campus, King Abdulaziz University, P.O. Box 80200, Jeddah 21589 , Saudi Arabia; wnhalharbe@kau.edu.sa; 11Nanotechnology Research Centre (NTRC), The British University in Egypt (BUE), Suez Desert Road, El-Sherouk City 11837, Cairo, Egypt; medahmed6@yahoo.com; 12Molecular Spectroscopy and Modeling Unit, Spectroscopy Department, National Research Centre, 33 El-Bohouth St., Dokki, Giza 12622, Egypt

**Keywords:** microstructure analysis, diffused reflectance UV-Vis, photocatalysis, tungsten trioxide, Methylene Blue and phenol degradations

## Abstract

The incorporation of tungsten trioxide (WO_3_) by various concentrations of graphitic carbon nitride (g-C_3_N_4_) was successfully studied. X-ray diffraction (XRD), Scanning Electron Microscope (SEM), and Diffused Reflectance UV-Vis techniques were applied to investigate morphological and microstructure analysis, diffused reflectance optical properties, and photocatalysis measurements of WO_3_/g-C_3_N_4_ photocatalyst composite organic compounds. The photocatalytic activity of incorporating WO3 into g-C_3_N_4_ composite organic compounds was evaluated by the photodegradation of both Methylene Blue (MB) dye and phenol under visible-light irradiation. Due to the high purity of the studied heterojunction composite series, no observed diffraction peaks appeared when incorporating WO_3_ into g-C_3_N_4_ composite organic compounds. The particle size of the prepared composite organic compound photocatalysts revealed no evident influence through the increase in WO_3_ atoms from the SEM characteristic. The direct and indirect bandgap were recorded for different mole ratios of WO_3_/g-C_3_N_4,_ and indicated no apparent impact on bandgap energy with increasing WO_3_ content in the composite photocatalyst. The composite photocatalysts’ properties better understand their photocatalytic activity degradations. The pseudo-first-order reaction constants (K) can be calculated by examining the kinetic photocatalytic activity.

## 1. Introduction

The photocatalytic performance of semiconductor materials, based on oxide semiconductors in their crystalline phases, has recently become desirable for producing hydrogen and oxygen via visible-light water splitting. Designing semiconductor composites produced between two semiconductor materials necessitates crystalline phase engineering [1,2,3]. The use of heterogeneous photocatalysis to convert water into hydrogen gas is regarded as one of the most promising options for addressing global energy and pollution problems [1,4,5]. In the present century, the rapid decline in energy sources, and increased power, waste, and renewable H_2_ energy sources, are attributable to worldwide demand for fossil fuels. A. Fujishima et al. studied for a long time how to establish a clean and renewable photocatalytic hydrogen-production method [6]. The absorption wavelength range of semiconductor materials such as oxides, sulfides, nitrides, and solid solutions was examined under visible light, to improve the absorption wavelength range [7,8].

Heterojunction photocatalysts were used extensively to improve the separation efficiency of photoexcited electron-hole pairs. A single-phase photocatalyst has significant limitations in a photocatalytic response, because photogenerated electrons and holes are quickly mixed [9] to develop optical properties. Jiang et al. doped phosphorus nanosheets with g-C3N4 and added carbon defects, which significantly increased the rate of hydrogen evolution through the photocatalytic method [9]. Furthermore, studies report an improved hydrogen production rate through using a mineral acid or phosphoric acid etching of g-C_3_N_4_ nanosheets to increase the number of active sites [10,11].

Graphite carbon nitride (g-C_3_N_4_) has recently gained new interest as a promising material in different applications such as photocatalysts, fuel cell electrodes, light-emitting devices, and chemical sensors. g-C_3_N_4_ stands out amongst many types of photocatalysts. A polymer photocatalyst bandgap is about 2.7 eV, allowing visible light to absorb up to 460 nm. Nevertheless, g-C_3_N_4_ can absorb visible light effectively since graphitic C_3_N_4_ has an adequate conduction band under illumination conditions that is able to be more harmful than protons formed by hydrogen [12,13]. Tungsten oxide (WO_3_) is considered a promising material [13,14]. It has been reported that synthesis of WO_3_/g-C_3_N_4_ composite organic compounds shapes a heterostructural composite photocatalyst [15,16],provided that the electron donor is an aqueous solution for triethanolamine. In the color-sensitization method, we looked at the composite catalyst’s hydrogen production behavior under visible-light irradiation. g-C_3_N_4_ composite, blended with WO_3_ using a planetary mill, was prepared using hydrothermal treatment to improve its photocatalytic activity [15]. The photocatalytic activity incorporated WO_3_ in g-C_3_N_4_ composite organic compounds [17]. J. Liang et al. [18] investigated the use of crystalline phase engineering in WO_3_/g-C_3_N_4_ composite organic compounds to improve photocatalytic activity under visible light. They demonstrated that WO_3_/g-C_3_N_4_ composites with h-WO_3_ show better dispersion of WO_3_, higher charge separation, and higher photocatalytic activity through visible-light photocatalytic degradation of Rhodamine B (RhB).

In this research, incorporating WO_3_ into g-C_3_N_4,_ composite organic compound photocatalysts were prepared, and exhibited improved photocatalytic activity under visible light. Their morphological, diffused reflectance UV-Vis optical, and (Methylene blue (MB) and phenol) photocatalytic activity properties were analyzed and discussed in detail, to understand the effects of WO_3_ on g-C_3_N_4_ composite organic compounds.

## 2. Experimental Parts

### 2.1. Regents 

In the experiment, an analytical grade of melamine, sodium tungstate dihydrate (Na_2_WO_4_·2H_2_O) ACS reagent, ≥99%, and other chemicals purchased from Sigma-Aldrich were used. These chemicals did not need further purification as they were of analytical grade. De-ionized water was utilized to avoid contamination.

### 2.2. Synthesis of Pure g-C_3_N_4_ and WO_3_/g-C_3_N_4_ Composite Organic Compounds

Simple g-C_3_N_4_ synthesis involves melamine pyrolysis prepared in an air atmosphere. Initially, 8 g of melamine was ground inside a crucible, then heated at a rate of 5 °C per min until it attained 550 °C, and then adjusted at this temperature for 2 h. Afterwards, the samples were left to cool down to ambient temperature. In the mortar, the resultant powder was then ground with a pestle. Following the first stage, Na_2_WO_4_·2H_2_O was added to melamine to synthesize the doped WO_3_/g-C_3_N_4_ composite organic compounds, after which the procedure for pure g-C_3_N_4_ was repeated. 

### 2.3. Characterization Techniques

The effect of WO_3_ on the physical structure of the g-C_3_N_4_ composite organic compounds for the 13 structures was characterized by X-ray diffraction. Shimadzu X-ray Diffractometer (XRD-6000 Series) was utilized with the standard copper X-ray tube at 30 kV and 30 mA with a wavelength equal to 1.5406 Å. The surface structure of all samples was evaluated by scanning electron microscopy (SEM-JSM6360 Series, with an acceleration voltage = 20 kV).

The effect of WO_3_ on g-C_3_N_4_ nanocomposite organic compounds was tested employing a UV-vis-NIR spectrophotometer model (Shimadzu UV-3600) with diffused reflectance in a wide range between 200 and 800 nm. The device was designed with the BaSO_4_ built-in sphere attachment as reference material. To counteract the WO_3_/g-C_3_N_4_ composites within the holder, as a guide for the BaSO_4_ and the other holder, a specifically constructed holder attached to the integrating sphere device was employed. The thickness of the holder corresponds to the thickness of the WO_3_/g-C_3_N_4_ nanocomposites. To track the photo-removal phase, a UV-visible spectrophotometer was used. 

The diffused reflectance was measured in ambient conditions using a JASCO UV-Vis-NIR-V-570 double beam spectrophotometer.

### 2.4. Photocatalytic Measurements

A simple wooden photoreactor tested the photocatalytic activity of WO3/g-C3N4 nanocomposites under visible-light spectrum radiation of samples, using both methylene blue (MB) and aqueous phenol as a pollutant example in wastewater. I.S. Yahia and his group designed the photoreactor in NLEBA, Ain Shams University (ASU), Egypt, consisting of two parts: the outer part was in the form of a box made from wood (height: 100 cm, width: 65 cm); the inner part had seven visible lamps (18 W, 60 cm length, 425 to 600 nm), and could be separately controlled. Each set of nanopowders was placed in a beaker containing 50 mL of MB (20 mg/L) (i.e., for the phenol photodegradation). The sample was magnetically stirred in the dark system to reach equilibrium between dye and photocatalyst. The sample was removed from the solution after a specific time (every 15 min), and the amount of sample remaining was then exposed to visible light. A UV-Vis spectrophotometer was used to analyze the samples.

## 3. Results and Discussions

### 3.1. Structural XRD Measurements

The crystalline structure phase was analyzed with the XRD technique to reveal the effect of doping WO_3_ in the final products. XRD patterns are shown in Figure 1 for pure g-C_3_N_4_ and its doping with WO_3_ on the g-C_3_N_4_ composite organic compounds, with various WO_3_ effects. As visualized in Figure 1, the g-C_3_N_4_ powder sample highlights two peaks at 2*θ* = 13.1°, and 2*θ* = 27.4°, which can be indexed to a small peak, (100) plane, pertinent to the in-plane structural packing of graphitic material, and a sharp peak, (002) plane, assigned to the interlayer spacing of the conjugated aromatic system. The (002) peak indicates a higher-density packing of the g-C_3_N_4_ molecules. This corresponds to the characteristic interplanar staking peaks of aromatic systems and the inter-layer structural packing, respectively [19,20]. These results are compared with Mo et al. [21]. The peak at *2θ* = 27.4° (JCPDS 87-1526) confirms the formation of hexagonal phase g-C_3_N_4_ powders [18,20,21]. The introduction of doping WO_3_ on g-C_3_N_4_ composite organic compounds reduces the maximum intensity of peaks. In Figure 1, we notice that only two peaks appear in the entirety of the prepared samples related to pure g-C_3_N_4_ except for the last sample, noted as 0.5 g WO_3_-doped g-C_3_N_4,_ showing the other two peaks related to WO_3_. The two peaks related to WO_3_ appear at *2θ* = 16.9° and 32.51°, related to (101) and (022), according to JCPDS 01-083-0950.

The average crystallite size for the prepared samples was measured using the Debye–Scherrer formula, as follows [22]:(1)D=0.94λ/βcosθ

The calculated crystallite size *D* depends on the broadening diffraction peak *β*, the diffraction angle θ, and the X-ray wavelength *λ.* Both dislocation density δ and lattice strain ε were calculated using the following equations [23,24]: (2)$$\delta =n/D^{2}$$
(3)ε=βcosθ/4

The XRD structural parameters for the prepared samples summarized in Table 1 show that the average crystallite size for the pure g-C_3_N_4_ is 39.17 nm, and this increases from 35.91, 35.93, 40.84, 44.83, to 81.67 nm for the 0.5, 0.1, 0.05, 0.01, and 0.001 WO_3_-doped g-C_3_N_4_ composites, respectively. 

The lower dislocation density values reflect the higher quality of the prepared samples.

### 3.2. Morphologies and Microstructure Analysis

To assess the effect of WO_3_ in g-C_3_N_4_ composite organic photocatalyst compounds, the morphology and microstructure of the pure g-C_3_N_4_ and its WO_3_-doped g-C_3_N_4_ composite organic compounds was studied by using SEM image analysis. Figure 2a–f presents the SEM images of the investigated composite photocatalysts. The morphology of pure g-C_3_N_4_ exhibited apparent granular aggregates and a typical sheet that consisted of small particles created from some irregular particles. Figure 2c–f shows that the WO_3_ particles were attached to the surface of the sheet g-C_3_N_4_. The particle sizes obtained from SEM images of the investigated composite organic compounds were gathered in Table 2. The particle sizes varied between 1.65 and 1.51 µm. As the WO_3_ doping particle contents increased, the g-C_3_N_4_ composite photocatalyst had no noticeable influence on the obtained particle size. A similar result is reported by X. Chu et al. [25].

### 3.3. Optical Properties of WO_3_ Doped g-C_3_N_4_ Nanocomposites

Figure 3a shows the optical UV-Vis diffused reflectance spectroscopy study to identify the bandgap energy of pure g-C_3_N_4_ and WO_3_-doped g-C_3_N_4_ composite organic photocatalyst compounds. The optical UV-Vis diffused reflectance spectra for other WO_3_/g-C_3_N_4_ photocatalyst composite samples should be the superimposed signals of g-C_3_N_4_. All diffused reflection spectra increased with increasing wavelengths up to 400 nm. The absorption edge spectra start to redshift towards the visible region, allocated to the intrinsic g-C_3_N_4_ bandgap [25].

The redshift suggested that the photocatalyst composite compounds could use sunlight and produce more electron-hole pairs, which would help the photocatalytic response. The above findings may be caused by the interactions between the WO_3_ and g-C_3_N_4_ incorporated into the heterojunction materials. Figure 3b,c displays the direct and indirect energy bandgap of pure g-C_3_N_4_ and its WO_3_ atom-doped g-C_3_N_4_ composite organic photocatalyst compounds, which can be estimated from the plots of (*ahυ*)*^2^* and (*ahυ*)^1/2^ vs. the incident photon energy (*hυ*) using Tauc’s formula [26]. The Kubelka–Munk function and its related absorption coefficient (*α*) are as follows [27,28,29]:*A**hυ* = (*F*(*R*) *h**υ*/*d*) = *A*(*h**υ*−*Eg*)*^r^*(4)
where *A is* a pre-factor, *E_g_* is the energy bandgap, and *r* limits the transition band type. When *r* = 2 is related to the indirect allowed band, *r* = 1/2 is associated with the direct allowed band. The bandgap of the photocatalyst was determined by Tauc’s plot, where *(αhυ)*^1/2^ with *hυ* axis (i.e., the linear portion of the plots *(αhυ)^2^* with *hυ* axis). By drawing a tangent line of each curve, each photocatalyst composite material’s bandgap was obtained from the intercept *hυ* axis. Therefore, the estimated bandgaps of pure g-C_3_N_4_ and its WO_3_ atom-doped g-C_3_N_4_ composite organic photocatalyst compounds, with various WO_3_ content, confirmed that the direct and indirect bandgaps for g-C_3_N_4_ were 2.83 [30] and 2.56 eV, respectively. This result was also reported by J.Y. Tai et al. [31]. For the indirect bandgap, we noted a slight change from 2.56 eV for the pure g-C_3_N_4_ changing to 2.6, 2.65, 2.66, 2.86, to 2.71 eV for the 0.5, 0.1, 0.05, 0.01, and 0.001 WO_3_-doped g-C_3_N_4_ composite, respectively. For the direct bandgap, a small change was also noted, from 2.83 eV for the pure g-C_3_N_4_ changing to 2.86. 2.88, 2.89, 2.90, 2.92 eV for the 0.5, 0.1, 0.05, 0.01, and 0.001 WO_3_-doped g-C_3_N_4_ composite, respectively. This indicated that the increase in WO_3_ content in the composite photocatalyst had no evident influence on bandgap energy. However, it was found that the movement and lifetime of the photogenerated charge carriers in g-C_3_N_4_ could be significantly affected by such a combination.

### 3.4. Kinetics of the Photodegradation Process of WO_3_-Doped g-C_3_N_4_ Nanocomposites

The photocatalytic performances of pure g-C_3_N_4_ and its WO_3_/g-C_3_N_4_ composites, with various concentrations, were assessed using MB and phenol contaminants under visible-light irradiation. The extent of the concentration of MB and phenol contaminants on the photocatalyst was measured in the dark system until equilibrium. The equilibrium between the prepared photocatalyst and the investigated organic pollutants, MB, as a colored dye, and phenol, as a colorless organic compound, was recognized after 30 min of stirring. The photocatalytic performances of pure g-C_3_N_4_ and its WO_3_/g-C_3_N_4_ composites with different photocatalysts were investigated for the photodegradation of MB and phenol contaminants. The achieved results are shown in Figure 4a,b. Figure 4a shows a limited photocatalytic performance of the pure g-C_3_N_4_ toward MB. However, the improved photocatalytic performance of WO_3_/g-C_3_N_4_ composites was recorded, indicating a major effect on the photocatalytic activity of g-C_3_N_4_ when modified with WO_3_. In general, the composite’s photocatalytic activity increased after WO_3_ was applied, and enhanced photocatalytic activity of 0.05% WO_3_ was noted. 

Nevertheless, the increase in WO_3_ (0.1 and 0.5) has been found to reduce photocatalytic efficiency, which may be due to the decrease in visible-light absorption catalytic sites. Additionally, the same behavior was observed for the degradation of phenol. The composite material containing 0.5% WO_3_ showed the maximum performance for phenol degradation, which can be attributed to the production of the WO_3_-semiconductor heterojunction effectively, which leads to transfer of charge from g-C_3_N_4_ under the visible-light irradiation. Moreover, the kinetics of the degradation of MB and phenol was noticed to follow the pseudo-first-order model, and the reaction rates were calculated by Equation (2) [32]:*ln*(*C_o_*/*C*) = *Kt*,(5)

Here, *C_o_* and *C* are the concentrations at the initial step and regular time intervals, respectively. Additionally, *K* is the pseudo-first-order reaction-rate constant (min^−1^) for investigated organic contaminants, and t is the time (min). Figure 5a,b shows the linear relationship between *ln*(*C_o_/C*) and *t*. According to these results, the photodegradation of MB and phenol pollutants follows pseudo-first-order reaction kinetics. Additionally, the degradation rate relies on the concentration of the organic substrate. Figure 6 shows the constant rate values for the degradation of MB and phenol contaminants. Percentage of degradation can be calculated to determine the efficiency of the prepared catalysts in the photocatalytic reaction, as illustrated in Figure 5. The efficiency of degradation increases with WO_3_ content to 0.5%, then the efficiency decreases. It can be noted that the photocatalytic degradation rates of MB had better rate-constant characteristics than the photocatalytic degradation rates of phenol. The photodegradation using MB and phenol contaminants in aqueous solution improvement onto pure g-C_3_N_4_ and its WO_3_/g-C_3_N_4_ photocatalyst composites with various concentrations was evaluated under visible light. Dependant on variations in the WO_3_ content, the plot of the irradiation time (*t*) against −*ln*(*C/C_o_*) was nearly a straight line. This result has also been reported by G. Lui et al. [33]. The corresponding estimated degradation rate constants (*K*) were added to Table 2.

### 3.5. Photodegradation Conduction Mechanism of WO_3_-Doped g-C_3_N_4_ Nanocomposites

The proposed mechanism for photocatalytic behavior of WO_3_/g-CN composite under the visible spectrum is interpreted in Figure 6. It is a successfully designed WO_3_/g-CN composite with various WO_3_. The Kubelka–Munk method was used to calculate the valence band (VB) and conduction band (CB) edge potentials of the prepared pure g-CN and WO_3_/g-CN with different concentrations of WO_3_. The direct and indirect bandgaps for g-CN are 2.83 and 2.56 eV, respectively. The increase in WO_3_ content in the WO_3_/g-CN composite significantly influences bandgap energy [34]. The proposed system suggests that both g-CN and WO_3_ will likely offer photogenerated charge carriers. The photogenerated electron in CB of WO_3_ then flows to VB of g-C_3_N_4_ due to electrostatic forces of attraction between the electron in CB of WO_3_ and the hole in VB g-CN, reducing the recombination of electrons and holes in the WO_3_/g-C_3_N_4_ composite and assisting in enhancing the charge separation spatially.

Furthermore, the electrons in the CB of g-C_3_N_4_ can also be retained by reducing the molecular oxygen O_2_ to form O^−2•^, due to the negative nature of the CB edge potential compared to the standard redox capability of O_2_/O^−2•^ [35]. Thus, the hole created in the VB of WO_3_ leads interacts with H_2_O to generate OH^•^ radicals, giving the more positive potential of the VB than the standard redox capability of OH/OH^−^. In this manner, the production of O^−2•^ and OH^•^ radicals plays a significant role in the photodegradation of MB and phenol degradation under visible light. With both OH^•^ and O^−2^, the phenol and Methylene Blue dyes produced CO_2_ and H_2_O [36,37].
WO_3_-g-C_3_N_4_ + hυ → e^−^ + h^+^(6)
O_2_ + e^−^ → O^−2•^(7)
H_2_O + h^+^ → OH^•^(8)
OH^•^ + O^−2•^ + organic dye (MB or Ph) → CO_2_+ H_2_O(9)

In comparison to the previous work, our 0.5% WO_3_/g-C_3_N_4_ has the highest photocatalytic activity of both MB and phenol in the presence of visible light, as shown in Table 3. Using the hydrothermal impregnation and calcination method, Cong Zhao et al. [38] obtained simple and inexpensive C-doped g-C_3_N_4_/WO_3_ as highly active photocatalysts. Tao Pan et al. [39] prepared WO_3_/g-C_3_N_4_ that showed improved photocatalytic activity for TC degradation. Perhaps most importantly, the UV-C-induced–TC degradation activity outperformed all other degradations regardless of concentration or pH level. According to Zhao et al. [40], the Ag/WO_2.9_/g-C_3_N_4_ composite outperforms pure g-C_3_N_4_ and pure WO_3_ in terms of light-selective adsorption and photocatalysis. Hong Yan et al. [41] proved that WO_3_/g-C3N4-modified nanocomposites had more significant photocatalytic activity than pure WO_3_ and pure g-C_3_N_4_. Junling Zhao et al. [19] discovered that WO_3_/g-C_3_N_4_ photocatalysts have increased degradation activities towards RhB dye when exposed to simulated sunlight. Minji Yoon et al. [42] demonstrated that WO_3_/g-C_3_N_4_ does not exhibit any photocatalytic activity on the degradation of p-nitrophenol. On the other hand, the degradation rate of p-nitrophenol was remarkably increased by the addition of Fenton to WO_3_/g-C_3_N_4_, which means that photocatalytic activity was further enhanced in the presence of hydrogen peroxide (H_2_O_2_) because of the generation of hydroxyl radicals.

### 3.6. Recycling of WO_3_-Doped g-C_3_N_4_ Nanocomposites

A 0.05 WO_3_/g-CN was subjected re-use in photodegradation of MB and phenol for 4 cycles. Figure 7 represents the recycling and reusability of WO_3_/g-C_3_N_4_. The samples exhibited high performance without further decreased reaction rate, which means that WO_3_/g-C_3_N_4_ is a promising material in MB and phenolic compound photodegradation.

## 4. Conclusions

Novel, efficient WO_3_/g-C_3_N_4_ nanocomposite with various concentrations of graphitic carbon nitride (g-C_3_N_4_) was prepared. The prepared composites exhibited improved photocatalytic activity on the degradation of MB and phenol under visible-light irradiation. The composite material containing 0.5% WO_3_ showed the maximum MB and phenol degradation performance. The kinetic study showed that the pseudo-first-order kinetic best fitted the photocatalytic process. A mechanism was proposed to degrade organic pollutants using WO_3_/g-C_3_N_4_ composites under visible-light irradiation.

## Figures and Tables

**Figure 1 materials-15-02482-f001:**
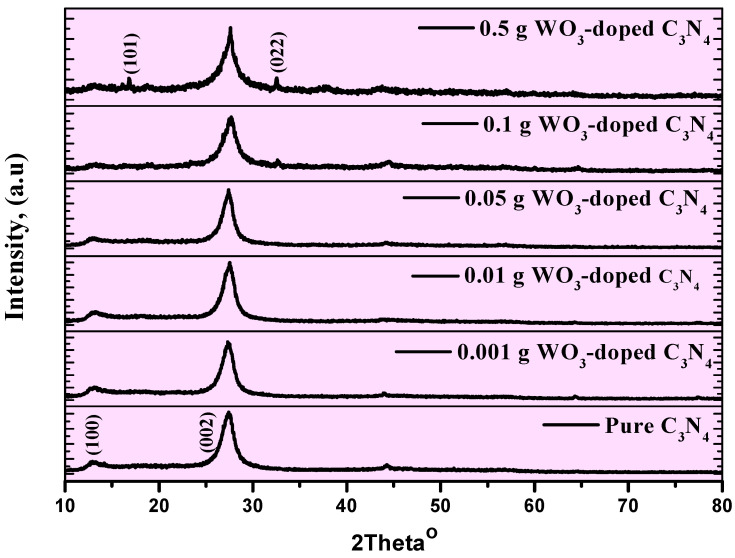
XRD patterns of pure g-C_3_N_4_ powder and its WO_3_ doping on g-C_3_N_4_ composite organic compounds (0.001, 0.01, 0.05, 0.1, and 0.5 g of WO_3_).

**Figure 2 materials-15-02482-f002:**
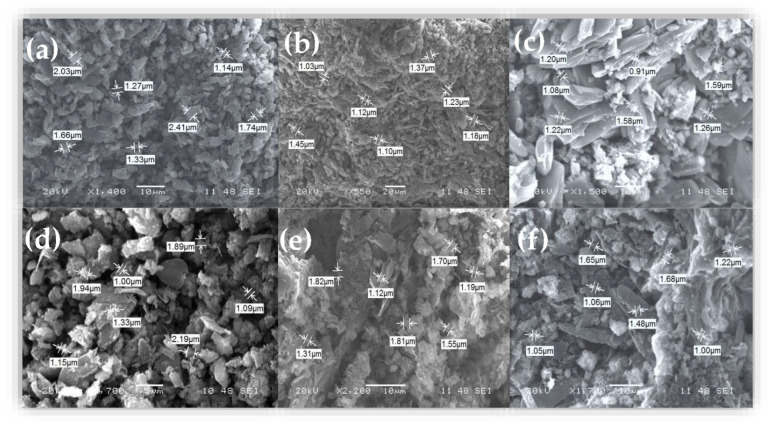
(**a**–**f**) SEM images of pure g-C_3_N_4_ and its WO_3_/g-C_3_N_4_ nanocomposite organic compounds with various amounts of tungsten oxide (0.001, 0.01, 0.05, 0.1, and 0.5 g of WO_3_, respectively).

**Figure 3 materials-15-02482-f003:**
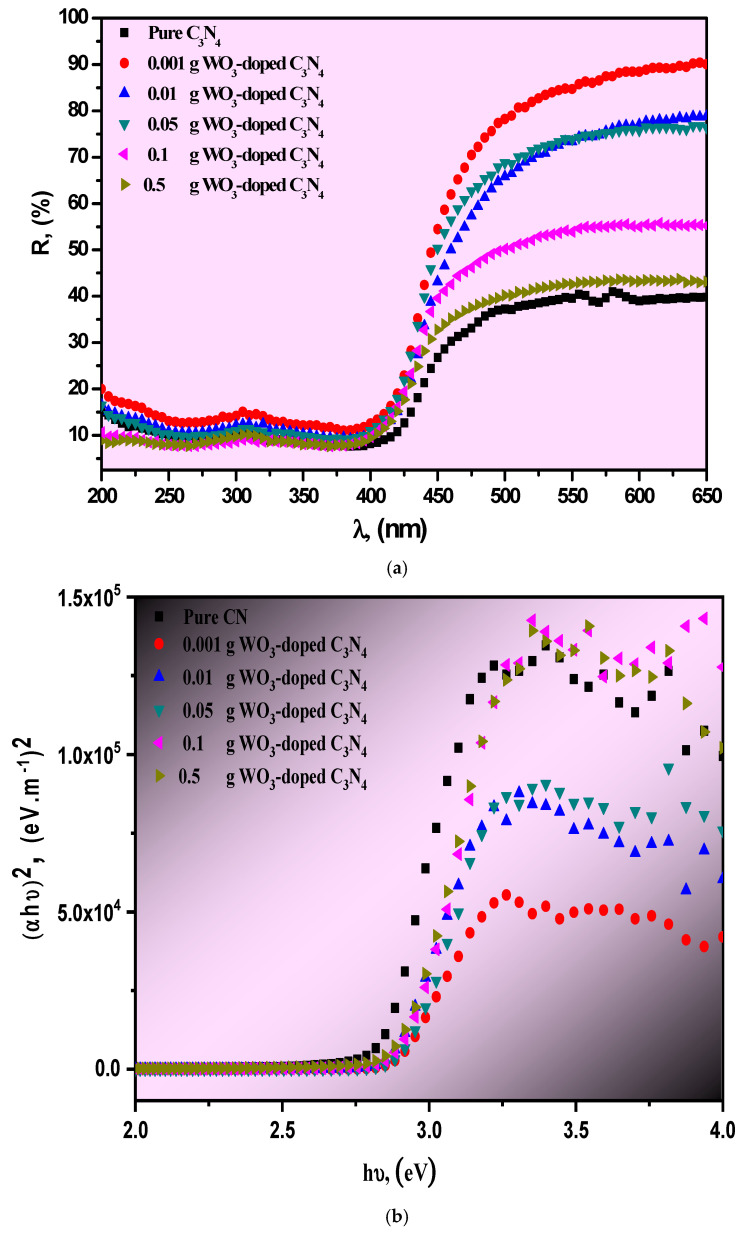
(**a**–**c**) Diffused reflectance optics UV-Vis (**a**), (*α**hυ*)*^2^* (**b**) and (*α**hυ*)*^1/2^* (**c**) versus the incident photon energy *h**υ* of pure g-C_3_N_4_ and its WO_3_/g-C_3_N_4,_ with various amounts of tungsten oxide (0.001, 0.01, 0.05, 0.1, and 0.5 g of WO_3_).

**Figure 4 materials-15-02482-f004:**
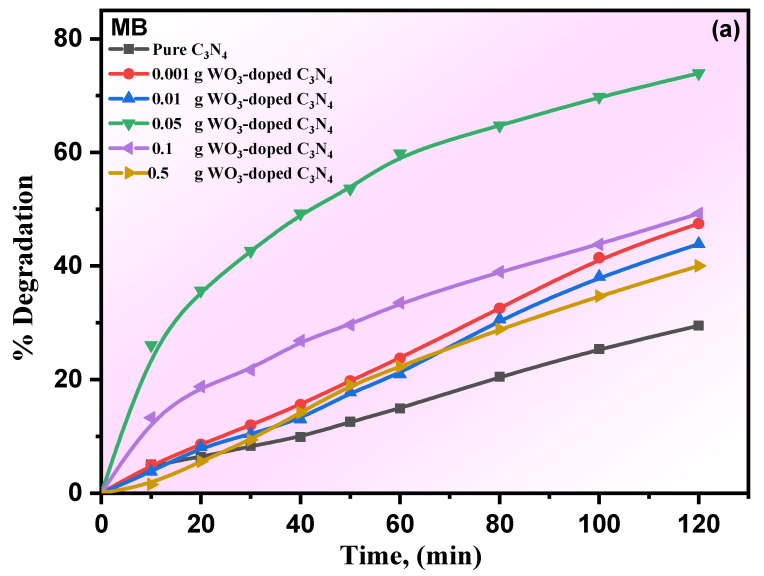
The degradation (%) of MB (**a**,**b**) phenol for pure g-C_3_N_4_ and its WO_3_/g-C_3_N_4_ nanocomposites.

**Figure 5 materials-15-02482-f005:**
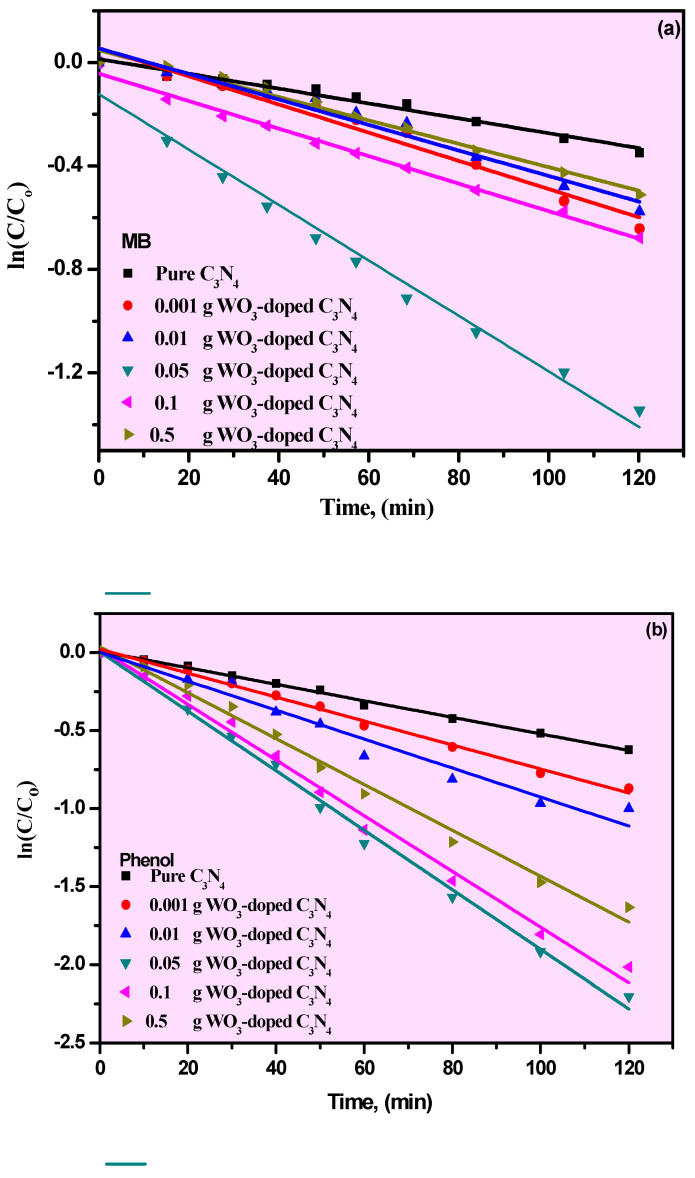
(**a**,**b**) The kinetic degradation curves of MB (**a**) and (**b**) Phenol for pure g-C_3_N_4_ and its WO_3_/g-C_3_N_4_ nanocomposites.

**Figure 6 materials-15-02482-f006:**
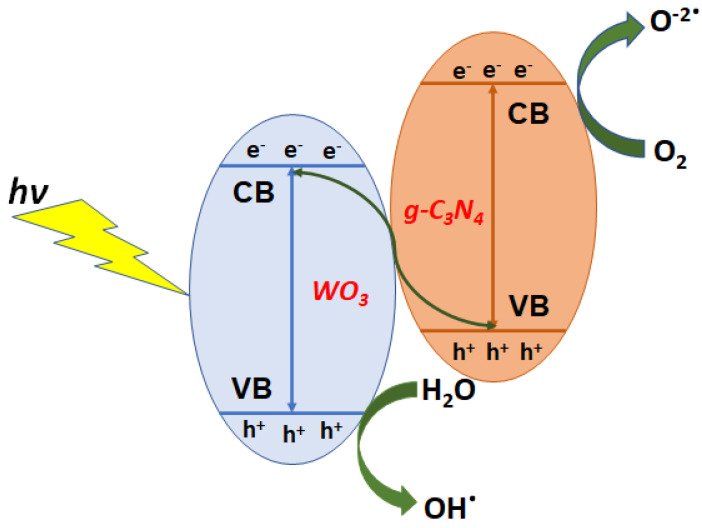
Photocatalytic mechanism of WO_3_/g-C_3_N_4_ nanocomposites.

**Figure 7 materials-15-02482-f007:**
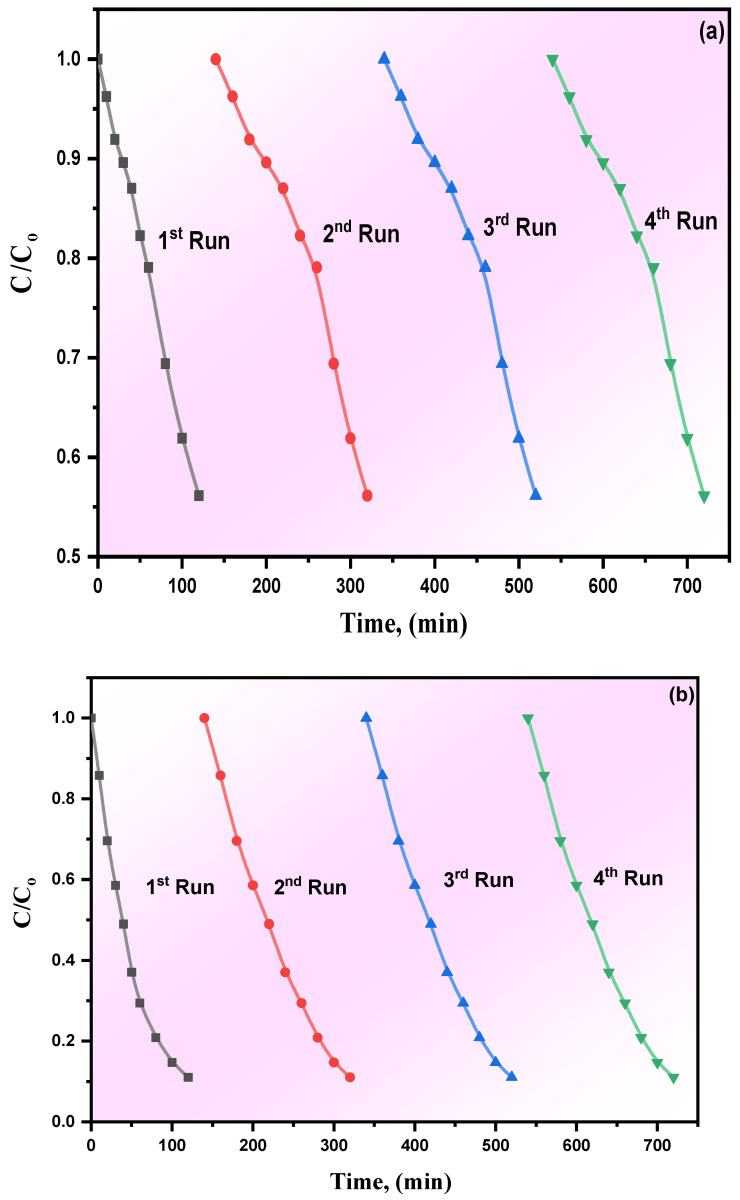
The recycling process for pure g-C_3_N_4_ and its WO_3_/g-C_3_N_4_ nanocomposites in photodegradation of (**a**) MB, (**b**) phenol.

**Table 1 materials-15-02482-t001:** N_4_ and its WO_3_/g-C_3_N_4_ with various amounts of tungsten oxide (0.001, 0.01, 0.05, 0.1, and 0.5% WO_3_).

Samples	Phases	Mean Values		
		Crystallite Size, (nm)	Dislocation Density, (1/(nm)^2^)	Lattice Strain
Pure g-C_3_N_4_	Pure g-C_3_N_4_	39.17	6.51 × 10^−4^	8.850 × 10^−4^
0.001 g WO_3_-doped g-C_3_N_4_	Pure g-C_3_N_4_	35.91	7.999 × 10^−4^	9.753 × 10^−4^
0.01 g WO_3_-doped g-C_3_N_4_	Pure g-C_3_N_4_	35.93	7.993 × 10^−4^	9.749 × 10^−4^
0.05 g WO_3_-doped g-C_3_N_4_	Pure g-C_3_N_4_	40.84	6.866 × 10^−4^	8.886 × 10^−4^
0.1 g WO_3_-doped g-C_3_N_4_	Pure g-C_3_N_4_	44.83	9.336 × 10^−4^	9.667 × 10^−4^
0.5 g WO_3_-doped g-C_3_N_4_	Phase 1: Pure g-C_3_N_4_	81.76	1.710 × 10^−4^	4.436 × 10^−4^
	Phase 2: WO_3_	52.93	3.641 × 10^−4^	6.593 × 10^−4^

**Table 2 materials-15-02482-t002:** Particle size and corresponding rate constants of pure g-C_3_N_4_ and its WO_3_/g-C_3_N_4,_ with various amounts of tungsten oxide (0.001, 0.01, 0.05, 0.1, and 0.5% WO_3_).

Samples	Particle Size, (µm)	*K*, (min^−1^)	
		MB	Phenol
Pure g-C_3_N_4_	1.65	0.0028	0.0053
0.001 g WO_3_/g-C_3_N_4_	1.21	0.0054	0.0064
0.01 g WO_3_/g-C_3_N_4_	1.26	0.0049	0.0092
0.05 g WO_3_/g-C_3_N_4_	1.3	0.0108	0.0194
0.1 g WO_3_/g-C_3_N_4_	1.5	0.0052	0.0172
0.5 g WO_3_/g-C_3_N_4_	1.51	0.0045	0.0145

**Table 3 materials-15-02482-t003:** Photocatalytic comparison between the prepared WO_3_/g-C_3_N_4_ and various previous works.

Photocatalyst	Method of Preparation	Organic Solution	Irradiation Time	Source	% Degradation	Refs.
WO_3_/g-CN	Pyrolysis	MB	120 min	Visible Light lamp	80%	Present work
		Phenol			90%	
C-doped g-C_3_N_4_/WO_3_	Hydrothermal impregnation	Tetracycline	60 min	UV-light lamp	75%	[33]
WO_3_/g-C_3_N_4_	Calcination method	Tetracycline	120 min	Xenon lamp	90.54%	[34]
Ag/WO_3_/g-C_3_N_4_	Calcination method	MB	300 min	UV-light lamp	-----	[35]
WO_3_/g-C_3_N_4_	Thermal polymerization	MO	120 min	Xenon lamp	93%	[36]
WO_3_/g-C_3_N_4_	Hydrothermal	RhB	90 min	UV-light lamp	91%	[37]
WO_3_/g-C_3_N_4_/ Photo-Fenton	Calcination method	P-nitrophenol	240 min	Xenon lamp	91%	[38]

## Data Availability

The data supporting this study’s findings are available from the corresponding author upon reasonable request.

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
