# Peer review of "Fabrication and Characterization of Highly Efficient As-Synthesized WO3/Graphitic-C3N4 Nanocomposite for Photocatalytic Degradation of Organic Compounds"

_materials, 2022, doi:10.3390/ma15072482_

Round 1

Reviewer 1 Report

In their paper, the authors have synthesized WO3/graphitic-C3N4 nanocomposite for photocatalytic degradation of both phenol and methylene blue.
Beore being accepted, some major issues should be taken into consideration as follows:

- In the XRD diffractogramms, there is a pic appaearing at 2θ = 33 for 0.5 and 1g  WO3 doped C3N4, it should be indexed!
-Another pic is appearing at 2θ = 17 for 0.5 WO3 doped C3N4
-Provide the standard XRD patterns for g-C3N4
-The crystallite size evolution as a function of the addition of WO3 should be calculated
- What is the % of doping
-From Fig 3(b) and 3(c), the composites show different band gap energies with slight differences, however, only one value is given in the text!
-The mechanism given in equations 3-8 is general, performing svengers' experiments would give an idea about the oxidizing species responsible on the degradation efficiency
-The photolysis tests (absence of catalyst) have not been shown!
-On table 2, add recycling runs of each catalyst 
-No explanation on the highest performance obtained for the composite with 0.05 g has been provided. The efficiency should be correlated to the caharcterization part
-Too many mistakes exist within the text , please revise ( i.e  separation efficiency of photogenerated Methylene Blue)
- Experimental results should be done in duplicates, if so, bar errors should be added

Author Response

Dear Reviewer, 

Thanks for your valuable comments; we tried to cover the whole comments.

We hope you find the manuscript suitable for publication in Materials and look forward to your reply. 

Reviewer 2 Report

The authors synthesized WO3/g-C3N4 nanocomposite and studied the photocatalytic degradation of methylene blue (MB) dye and phenol under visible light irradiation. The mechanism for the enhanced photocatalytic degradation properties were studied. The manuscript can be accepted for publication after addressing the following points: 1. According to Figure 1, some new XRD peaks appeared when increased the content of WO3, indicating that WO3/g-C3N4 is a composite. Thus, it is not suitable to use “WO3-doped C3N4”. The authors should correct this to avoid possible misunderstanding. In addition, the XRD peaks of WO3 and C3N4 should be clearly labelled. 2. In Figure 2, the SEM images should be labelled with “a-f”. 3. Figure 7 seems incorrect. The authors should also measure the bandgap and band edge positions of WO3, so the band alignment between C3N4 and WO3 can be correctly illustrated. 4. In the whole manuscript, the numbers in WO3 and C3N4 should be subscripted. 5. Some relevant literature should be discussed and cited. For example: J. Mater. Sci. Technol., 2022, 104, 155-162; J. Cent. South Univ., 2021, 28, 2345-2359.

Author Response

(The authors gave the same response as above.)

Reviewer 3 Report

This work reported the fabrication and characterization of WO3/graphitic-C3N4 nanocomposite for photocatalytic 3 degradation of organic compounds methylene blue (MB) dye and phenol under visible light irradiation. This manuscript needs to be revised before the acceptance for publication in this journal.

  1. The statement “Due to the high purity of the studied heterojunction composite series, no observed diffraction peaks appear when incorporating WO3 into g-C3N4 composite organic compounds” is confusing.
  2. (1) About the statement “The direct and indirect bandgap were recorded for different mole ratios of WO3/g-C3N4” and Figure 3b-c, the bandgap characteristics (direct or indirect bandgap) should be clarified in advance based on the intrinsic features of semiconductors and then the values of pre-factor (r) could be determined. (2) Please explain change trend of Diffused reflectance optics UV-Vis in Figure 3a with the increase of WO3 content. (3) the data of Figure 3b-c seems not consistent with that of Figure 3a.
  3. The description “8gm of urea has been ground inside a crucible” is confusing. Please confirm the unit “gm” as well as the precursor “urea” since “melamine” was mentioned in the context.
  4. Please give the mass ratio, instead of the mass (0.001, 0.01, 0.05, 0.1, and 0.5 g of WO3), of WO3 to g-C3N4 for better presentation.
  5. “Particle size” in Table 1 seems not meaningful. Please carry out the characterization of porosity properties to get the specific surface areas, pore size distribution and pore volumes (refer to: Chemical Engineering Journal, 2022, 431, 134101).
  6. Figure 4 should be shown after Figure 5 and was similar (repetitive) to Figure 6.
  7. For Figure 7, “W” should be “WO3” and WO3 also could be excited by visible light. Besides, the band alignment of g-C3N4 and WO3 should be given and the charge transfer between g-C3N4 and WO3 should be denoted (refer to: Journal of Colloid & Interface Science, 2022, 608, 2058-2065).

Author Response

(The authors gave the same response as above.)

Round 2

Reviewer 1 Report

The authors have answered the asked comments and suggestions

The revised MS can be accepted for publication

Reviewer 2 Report

The authors have addressed all comments from reviewers. The manuscript can be accepted at the current version.

This manuscript is a resubmission of an earlier submission. The following is a list of the peer review reports and author responses from that submission.